# Informal Caregiving in Adolescents from 10 to 16 Years Old: A Longitudinal Study Using Data from the Tokyo Teen Cohort

**DOI:** 10.3390/ijerph20156482

**Published:** 2023-07-31

**Authors:** Miharu Nakanishi, Daniel Stanyon, Marcus Richards, Syudo Yamasaki, Shuntaro Ando, Kaori Endo, Mariko Hosozawa, Mitsuhiro Miyashita, Mariko Hiraiwa-Hasegawa, Kiyoto Kasai, Atsushi Nishida

**Affiliations:** 1Department of Public Health and Primary Care, Leiden University Medical Center, 2300RC Leiden, The Netherlands; 2Department of Psychiatric Nursing, Tohoku University Graduate School of Medicine, Sendai-shi 980-8575, Japan; 3Mental Health Promotion Unit, Research Center for Social Science & Medicine, Tokyo Metropolitan Institute of Medical Science, Setagaya-ku, Tokyo 156-8506, Japan; 4MRC Unit for Lifelong Health & Ageing at UCL, University College London, London SW1H 9NA, UK; 5Department of Neuropsychiatry, Graduate School of Medicine, The University of Tokyo, Bunkyo-ku, Tokyo 113-8655, Japan; 6Institute for Global Health Policy Research, National Center for Global Health and Medicine, Shinjuku-ku, Tokyo 162-8655, Japan; 7Department of Evolutionary Studies of Biosystems, The Graduate University for the Advanced Studies, Hayama 240-0193, Japan; 8The International Research Center for Neurointelligence (WPI-IRCN), The University of Tokyo Institutes for Advanced Study (UTIAS), Bunkyo-ku, Tokyo 113-0033, Japan

**Keywords:** adolescent, family caregiver, informal care, young caregivers

## Abstract

There is growing evidence of the impact of informal caregiving on adolescent mental health, and its role is often hidden unintentionally or intentionally, which may hamper early identification and support for young informal caregivers. However, the quantitative evidence regarding household factors relating to informal caregiving has mostly been based on cross-sectional findings. This study examines the longitudinal associations between household characteristics and the duration of informal caregiving in adolescents from 10 to 16 years of age. Child–household respondent pairs (*n* = 2331) from the Tokyo Teen Cohort in Japan were followed every 2 years from 10 to 16 years of age. Informal caregiving was assessed repeatedly based on the household respondent’s survey responses. Persistent caregiving was defined as daily caregiving at two or more waves. There were 2.2% of children who gave daily care at two or more waves. Cross-sectional associations with daily informal caregiving at each wave were found with girls, low household income, and cohabiting with grandparents. A significant association with persistent caregiving was found only in cohabiting with grandparents at 10 years of age after adjusting for sex, number of siblings, single parent, and household income. Our longitudinal examination highlighted cohabiting with grandparents as a preceding factor for persistent caregiving. Identification and support for young informal caregivers should be integrated into social care service systems for older adults. The mechanism of persistent caregiving requires clarification.

## 1. Introduction

### 1.1. Informal Caregiving among Children and Adolescents

Children and adolescents who provide unpaid care to other family members are more likely to report negative mental health and well-being than those without caregiving responsibilities. Young caregivers are defined as individuals under 18 years of age who provide unpaid care to family members with disabilities, chronic illnesses, mental health issues, and substance use problems [1]. They are assumed to have substantial personal and emotional caregiving responsibilities for family members who need help and in the management of household tasks [1,2]. Young caregivers comprise 2–8% of the population of their age group in England [3,4], 12% in Scotland [5], 14% in France [6], and 6–7% in Japan [7,8,9,10,11]. Adolescence is a period of significant psychological vulnerability in social and emotional development [12]. The provision of unpaid care to family members during this period affects the caregiver’s identity development, social integration, and interactions with peers [2], which could result in worse mental and psychosocial health outcomes [5,13,14]. Nonetheless, the challenges faced by young caregivers have been under-recognized in law and services in several countries [4,15].

### 1.2. Challenges in the Identification of Young Informal Caregivers

Supporting young informal caregivers can be hampered by the challenges surrounding the identification of them in their school management systems. Young people often hide their roles as they are concerned about any stigma surrounding their family, especially if their family is one with mental health and addiction problems [2,3]. Signs that someone might have caring responsibilities include absenteeism, lateness, incomplete homework, being a victim of bullying, and disruptive behavior [4]. These signs imply that such young caregivers have already faced several disadvantages due to their overwhelming caring responsibilities. Understanding the preceding factors related to informal caregiving will provide implications for the earlier identification and linkage of young informal caregivers with available support to mitigate the negative impact on their social lives. Some household characteristics, such as single-parent households, low-income households, and households with many siblings have been suggested [4,16]. Social protection for children and families with such disadvantaged socioeconomic backgrounds can play a role in identification and support for young informal caregivers. However, the quantitative evidence regarding informal caregiving thus far has been mostly based on cross-sectional findings [17,18]. While some studies using longitudinal data have repeatedly assessed mental health outcomes [14,19,20], information on informal caregiving has often been evaluated at one time point. A longer duration of time as a young informal caregiver can result in negative impacts on mental health outcomes [2]; hence, priority should be given to identifying how caregiving responsibilities become persistent in childhood. Despite growing evidence showing the experiences and outcomes of young informal caregivers, the lack of identification may lead to delayed or missed support delivery through key stages of childhood/adolescence. Hence, there is an urgent need for the early identification of young informal caregiving.

### 1.3. Tokyo Teen Cohort (TTC)

In Japan, the first nationwide survey on the prevalence of young informal caregiving was administered in 2020 [10]. Yet, there has been no national legislation for identification and support for young informal caregivers. The Tokyo metropolitan region has the largest number of children across 47 regions [21]. The number of children increased in Tokyo over a 5-year period from 2015 to 2020, despite a declining trend in all other 46 regions [21]. The Tokyo Teen Cohort (TTC) is a prospective population-based birth cohort study aimed at investigating physiological and psychological development in adolescence [22]. Population-based sampling from inhabitants in metropolitan areas enables TTC to include a large number of children in the cohort using a prospective design.

### 1.4. Aim of the Study

This study aimed to investigate household factors relating to persistent informal caregiving in adolescents. We used data from TTC, which assessed informal caregiving every 2 years. This enabled us to group participants based on longitudinal frequencies of caregiving. Additionally, our longitudinal examination can also illuminate preceding factors by prioritizing considerations for persistent caregiving. Specifically, we featured cohabiting with grandparents in addition to sex, number of siblings, single parent, and low annual household income among the factors assumed to be associated with informal caregiving [14]. Our findings will provide us with an understanding of the duration of informal caregiving in adolescence and the household factors for the early identification of young informal caregivers.

## 2. Materials and Methods

### 2.1. Study Design

A longitudinal design was adopted for this study.

### 2.2. Setting

The TTC is an ongoing population-based cohort study following the physiological and psychological development of 3171 children born between 2002 and 2004 residing in three municipalities in the metropolitan area of Tokyo, Japan. The sample was recruited from participants in the Tokyo Early Adolescence Survey (T-EAS), a cross-sectional survey on the psychological and physical development of 10-year-old children [22]; data from T-EAS were treated as the first wave of data for the TTC. Children who lived in three municipalities (Setagaya-ku, Mitaka-shi, and Chofu-shi) and were born between September 2002 and August 2004 were recruited using the resident register in each municipality. Of the 14,553 randomly chosen children, 4319 could not be contacted. Of the 10,234 children who were accessible, 4478 agreed to participate in T-EAS (response rate: 43.8%). Participants of TTC were chosen from 4478 children who participated in T-EAS. An oversampling method was used, considering the low follow-up rate of families with a low annual household income. A detailed description of the TTC is available [22]. A trained interviewer visited the participants’ homes to administer self-report questionnaires during each survey. The first-wave survey was conducted between October 2012 and January 2015, when the children were 10 years old. Pairs of child and primary household respondents were followed every 2 years.

Paper questionnaires were used for the first- to third-wave surveys. In the fourth-wave survey during the COVID-19 pandemic, online questionnaires were adapted for data collection. Participating children and household respondents took 60 min to complete the questionnaire during each wave.

### 2.3. Participants

The eligible sample was defined as households that provided information on informal caregiving for all fourth-wave surveys from 10 to 16 years of age. From 2616 households participating in the fourth-wave survey, 2331 children and primary household respondents were included in the analysis (Figure 1). The 2331 participants included had a lower proportion of single-parent households and households cohabiting with grandparents than the 840 excluded or drop-out participants (Appendix A).

Written informed consent was obtained from parents prior to participation in each wave of data collection. All study procedures were approved by the institutional review boards of the relevant institutions.

### 2.4. Measurements

The survey was completed during two home visits in each wave. During the first visit, written informed consent was obtained from the primary household respondent (generally the mother), and the Part 1 self-report questionnaires were distributed. Participants were asked to complete the questionnaires at home, before the second visit. During the second visit, the adolescents and respondents were asked to complete the Part 2 self-report questionnaires separately. The questionnaires were enclosed in envelopes immediately after their completion. Additionally, the respondents underwent semi-structured interviews. All data were collected anonymously.

The second-wave survey was conducted when the children were 12 years old, between July 2014 and January 2017. The third-wave survey, when the children were 14 years old, was conducted between October 2016 and January 2019. The fourth-wave survey, when the children were 16 years old, was conducted between February 2019 and September 2021.

Table 1 shows the measurements used in this study. All information, including that of informal caregiving, was provided by primary household respondents.

### 2.5. Informal Caregiving

Primary household respondents were asked, “How often does your child take care of someone in the household who is an older adult or has an illness or physical disability?”, with valid response options of “everyday”, “sometimes a week”, “about once a week”, “once a month”, “less than once a month”, and “never (no one in need of care)” (Table 1). Given that the effect of caregiving on mental health may vary according to caregiving intensity [14], we used a binary variable for our analysis: daily (everyday) caregiving vs. less than daily caregiving or no caregiving.

As for the duration of informal caregiving, we calculated how many times daily informal caregiving was reported across the four survey waves from 10 to 16 years of age. Most participants (91.7%) provided no daily caregiving; 6.1% experienced daily caregiving at one wave and 2.2% experienced daily caregiving at two or more waves (Figure 2). Therefore, we created a three-category variable based on number of times of daily caregiving: never, occasional (once), and persistent (twice or more) caregiving. The definition of persistent caregiving was based on a previous study on young adult caregivers in the United Kingdom [16].

### 2.6. Household Characteristics

We measured household characteristics that were previously assumed to be associated with informal caregiving [14], including number of siblings, single parent, and low annual household income. The child’s sex was also used in the analysis as girls are more likely to be selected to be informal caregivers [2]. In addition to these variables suggested by the literature, we included cohabiting with grandparents among our household characteristics as a potential factor in informal caregiving. In Japan, 20.7% of frail older adults in 2019 lived with their adult children, who cared for their parents as primary informal caregivers [23].

The number of siblings and those cohabiting with grandparents were identified based on responses regarding household composition. Primary household respondents were asked to describe all household members’ sex, age, and relationship (i.e., brother, sister, mother, father, grandmother, grandfather) with the participating children. The number of siblings was calculated by summing the number of brothers and sisters. Cohabiting with grandparents was coded with binary categories: “yes” (one or more grandparents in the household) or “no”. Single parent was identified based on the primary household respondent’s response about having a partner or not. Annual household income was assessed using four categories: “less than four million yen”, “4–6.99 million yen”, “7–9.99 million yen”, and “10 or more million yen”. Responses were re-classified into binary categories, “less than four million yen” and “four million yen or more”, to perform multivariate analyses.

Data on household characteristics were repeatedly collected in each survey wave, except for data on number of siblings and cohabiting with grandparents. Number of siblings was only collected in the first-wave survey at 10 years of age. Information on cohabiting with grandparents was collected from the first- to third-wave surveys, but not in the fourth-wave survey at 16 years of age.

### 2.7. Statistical Analysis

The characteristics of the sample at 10 years of age were described based on their duration of daily informal caregiving.

To examine whether the frequency of informal caregiving changed over time (age), a Kruskal-Wallis H test was performed between the four times of assessment.

To investigate the cross-sectional associations between daily informal caregiving and household characteristics, a multivariate binomial regression analysis was performed at each survey wave. A dependent variable examined was daily informal caregiving vs. less than daily or no caregiving. Independent variables included all covariates at each wave, including the child’s sex, number of siblings, single parent, low annual household income, and cohabiting with grandparents.

To test the longitudinal associations between persistent caregiving and household characteristics, a multivariate multinomial regression analysis was also conducted. Dependent variables comprised occasional and persistent caregiving with reference to a never-caregiving group from 10 to 16 years of age. Independent variables included all covariates comprising the child’s sex, number of siblings, single parent, low annual household income, and cohabiting with grandparents when the child was 10 years old.

In these regression analyses, the full information maximum likelihood was used to handle missing data [24]. Data management was conducted using Stata version 17.0 (StataCorp). Regression analyses were performed using Mplus 8.8 (Muthen & Muthen). Statistical significance was set at *p* < 0.05.

### 2.8. Sensitivity Analysis

We reanalyzed the multivariate models by excluding individuals with missing data.

## 3. Results

### 3.1. Characteristics of Caregivers

There were 275 (11.8%) caregivers at 10 years of age, 318 (13.6%) at 12 years, 313 (13.4%) at 14 years, and 367 (15.7%) at 16 years. At each survey wave, around 3.0% of children were reported to provide daily informal care to family members (Table 2). The distribution of frequency of caregiving did not differ across the four-wave survey (Kruskal-Wallis H test, χ^2^(3) = 5.21, *p* = 0.157).

Of the total 2331 participants, 46.6% were girls, 82.3% had siblings, and 4.2% lived in single-parent households. There were 9.9% who lived in households with a low annual income (less than 400 million yen). There were 8.2% cohabiting with their grandparents (Table 3).

### 3.2. Cross-Sectional Association between Daily Informal Caregiving and Household Characteristics

In the multivariate binomial logistic regression model, girls were more likely to be engaged in daily informal caregiving at all times of assessment, except at 14 years of age. Children cohabiting with their grandparents were more likely to be engaged in daily informal caregiving at all times of assessment. At 14 years, children in households with a low annual income were more likely to be engaged in daily informal caregiving (Table 4). Number of siblings and single-parent household did not show associations with daily informal caregiving at any time of assessment.

### 3.3. Longitudinal Association between Duration of Daily Informal Caregiving and Household Characteristics

In the multivariate multinomial logistic regression model, children cohabiting with grandparents were more likely to be engaged both in persistent and occasional caregiving (Table 5). The child’s sex, number of siblings, single-parent household, and low annual household income did not show associations with persistent or occasional caregiving.

### 3.4. Sensitivity Analysis

In the sensitivity analysis that excluded individuals with missing data, cross-sectional associations did not significantly change for daily caregiving, sex, low household income, or cohabiting with grandparents (Appendix A).

The sensitivity analysis of longitudinal associations did not alter associations between persistent or occasional daily caregiving and cohabiting with grandparents (Appendix A).

## 4. Discussion

This large population-based study investigated the duration of daily informal caregiving among Japanese children from 10 to 16 years of age. The prevalence of daily informal caregiving at each wave was around 3% and was stable across 10- to 16-year-olds; there was no significant change in frequency over time. Persistent caregiving, defined as daily caregiving at two or more waves in this study, was observed in 2% of the participants. Cohabiting with grandparents demonstrated consistent associations with daily informal caregiving at each wave and persistent caregiving. The results were robust across the missing data estimation models and complete case analysis. Our longitudinal examination highlights cohabiting with grandparents as a preceding factor for persistent caregiving.

Our study is the first to indicate that cohabiting with grandparents is associated with temporary and long-term informal caregiving among children. Some children may have provided informal care to their grandparents who were frail older adults who needed care and support, as suggested by findings from European countries [3,25,26,27]. Caring for older adults could be characterized by a potentially longer duration due to a gradual and progressive decline in physical health. In Japan, individuals aged 65 years or older are mandatorily insured by the public long-term care insurance program. Users of in-home services are assigned care managers who handle monthly care plans under the program [28]. However, such key stakeholders in social care for older adults may not fully acknowledge young informal caregiving in the household. Identification and support for young informal caregivers should be integrated into social care service systems for older adults to build upon stakeholders’ abilities. It should be noted that our study did not collect information on the person who was cared for by the child. Therefore, cohabiting with grandparents may not have necessarily represented intergenerational caring. For example, parents of children may have carried out caring for their own parents who were frail older adults, delegating care for siblings and household chores to children. Cohabiting within three generations could also indicate parents’ challenges regarding finances and health that necessitate support from grandparents as well as informal care from their children. Further research is needed to clarify the mechanism of how temporary informal caregiving occurs and evolves into persistent care responsibilities.

Cross-sectional associations between girls and temporary informal caregiving were consistent with previous studies [2]. Globally, girls and young women are relied on as a caring resource [29]. The unequal care responsibilities between women and men could shape a sense of care obligations among girls in case of the onset or rise of care needs for family members in the household. Their caring activities consequently reduce their labor force participation, education, and income, resulting in expanded inequalities [18,30,31]. Social protection for young informal caregivers should address gender inequality.

### Strengths and Limitations

This is the first study to quantitatively examine the associations between persistent informal caregiving and household characteristics. The strength of our study lies in the use of a large representative dataset. The dataset provided repeated measurements of informal caregiving using the same scales across different survey waves. Furthermore, caregiving activities were reported by the primary household respondents (i.e., parents), which may have reduced the self-reporting bias associated with the underestimation of caregiving. The implications of this study include marking cohabiting with grandparents as a potential signal of young informal caregiving for social care sectors. Future studies that identify the mechanism of persistent caregiving in cohabiting households would provide insights into risk factors and prevention. Further policies, regulations, and educational efforts should be explored to increase gatekeeping abilities in social care service systems for older adults.

Regarding limitations, our results are based on Japanese individuals; therefore, they cannot be generalized to populations in other countries. While the prevalence of daily informal caregiving (3% of the population) in this study was similar to that previously reported in Japan [10,11], it appeared to be relatively lower than that of Australia [14]. Although the instruments used in TTC have been developed with input from an international advisory board [22], the consistency between paper and online administration should be considered with caution. Furthermore, 26.5% of initial participants at baseline were excluded from our analysis. The excluded participants had more children in single-parent households and households with a low annual income. Therefore, the associations between these variables and informal caregiving might have been underestimated.

## 5. Conclusions

Using a large population-based dataset coupled with quantitative methods, this study demonstrated the consistent associations between cohabiting with grandparents and daily caregiving. Furthermore, the longitudinal association implied that cohabiting with grandparents may be a preceding factor for persistent caregiving. The results highlight the importance of the ability to identify and support young informal caregivers as stakeholders of social care service systems for older adults.

## Figures and Tables

**Figure 1 ijerph-20-06482-f001:**
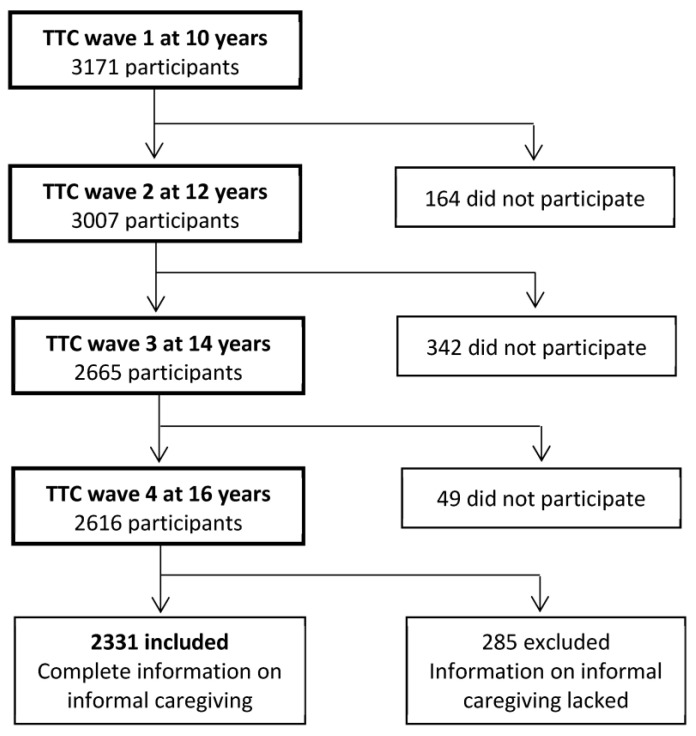
Flow chart of the study.

**Figure 2 ijerph-20-06482-f002:**
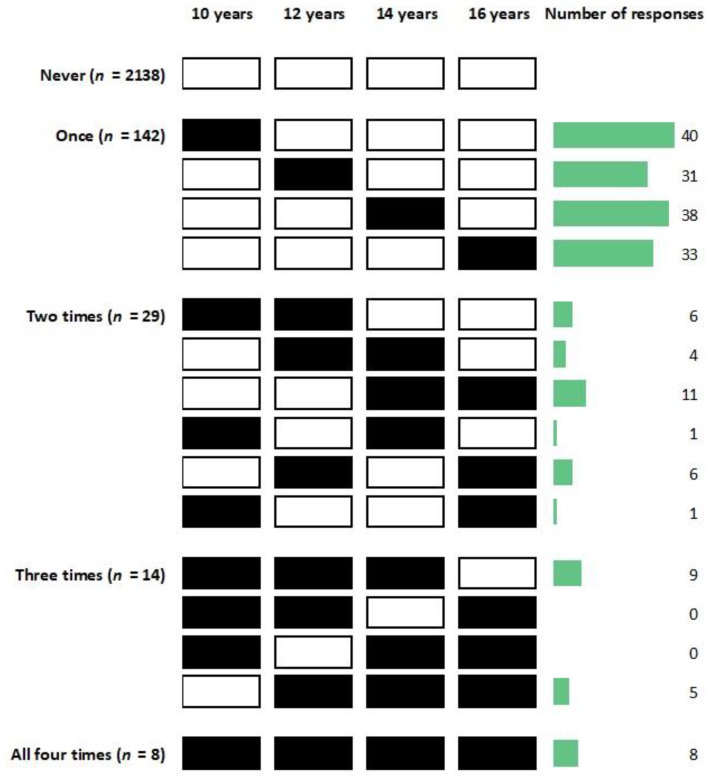
Number of times of daily informal caregiving reported across four waves.

**Table 1 ijerph-20-06482-t001:** Measurements used in this study.

	Wave 1	Wave 2	Wave 3	Wave 4
Informal caregiving	X	X	X	X
Sex	X			
Household characteristics				
Number of siblings	X			
Single parent	X	X	X	X
Annual household income	X	X	X	X
Cohabiting with grandparents	X	X	X	

X: Answered by primary household respondents.

**Table 2 ijerph-20-06482-t002:** Informal caregiving status across four-wave survey.

	10 Years	12 Years	14 Years	16 Years
No caregiving	2056 (88.2)	2013 (86.4)	2018 (86.6)	1964 (84.3)
Less than daily caregiving	210 (9.0)	249 (10.7)	237 (10.2)	303 (13.0)
Less than once a month	50 (2.1)	62 (2.7)	66 (2.8)	58 (2.5)
Once a month	59 (2.5)	76 (3.3)	78 (3.3)	109 (4.7)
About once a week	37 (1.6)	36 (1.5)	37 (1.6)	52 (2.2)
A few times a week	64 (2.7)	75 (3.2)	56 (2.4)	84 (3.6)
Daily caregiving (everyday)	65 (2.8)	69 (3.0)	76 (3.3)	64 (2.7)

Numbers and percentages per 2331 children.

**Table 3 ijerph-20-06482-t003:** Characteristics of participants when the child was 10 years old by duration of daily informal caregiving from 10 to 16 years of age.

	Total (*n* = 2331)	Never (*n* = 2138)	Occasional ^1^ (*n* = 142)	Persistent ^2^ (*n* = 51)
Sex, N (%)				
Girl	1087 (46.6)	982 (45.9)	75 (52.8)	30 (58.8)
Boy	1244 (53.4)	1156 (54.1)	67 (47.2)	21 (41.2)
Number of siblings, N (%)				
None	413 (17.7)	377 (17.6)	27 (19.0)	9 (17.6)
One	1335 (57.3)	1236 (57.8)	74 (52.1)	25 (49.0)
Two or more	583 (25.0)	525 (24.6)	41 (28.9)	17 (33.3)
Single parent, N (%)				
Yes	99 (4.2)	86 (4.0)	7 (4.9)	6 (11.8)
No	2232 (95.8)	2052 (96.0)	135 (95.1)	45 (88.2)
Annual household income, N (%) *				
Less than 4 million yen	222 (9.9)	195 (9.4)	18 (13.2)	9 (18.4)
4–6.99 million yen	653 (29.0)	592 (28.7)	48 (35.3)	13 (26.5)
7–9.99 million yen	682 (30.3)	633 (30.7)	38 (27.9)	11 (22.4)
10 or more million yen	692 (30.8)	644 (31.2)	32 (23.5)	16 (32.7)
Cohabiting with grandparents, N (%)				
Yes	192 (8.2)	137 (6.4)	35 (24.6)	20 (39.2)
No	2139 (91.8)	2001 (93.6)	107 (75.4)	31 (60.8)

^1^ “Occasional” group had daily informal caregiving at one wave from 10 to 16 years of age. ^2^ “Persistent” group had daily informal caregiving at two or more waves from 10 to 16 years of age. * There were 82 participants who did not answer the question, 74 never provided care, 6 who occasionally provided care, and 2 who engaged in persistent daily informal caregiving.

**Table 4 ijerph-20-06482-t004:** Cross-sectional association between daily informal caregiving and household characteristics at each wave.

	10 Years	12 Years	14 Years	16 Years
Sex, girl	1.959 (1.170–3.280) *	1.798 (1.099–2.957) *	1.206 (0.750–1.939)	1.806 (1.075–3.032) *
Number of siblings ^1^	1.171 (0.877–1.564)	1.281 (0.969–1.695)	1.050 (0.792–1.392)	1.255 (0.941–1.675)
Single parent	0.727 (0.201–2.621)	2.087 (0.817–5.329)	1.590 (0.707–3.575)	0.948 (0.346–2.597)
Low annual household income ^2^	1.208 (0.532–2.740)	1.266 (0.544–2.946)	2.980 (1.432–6.201) **	2.023 (0.844–4.850)
Cohabiting with grandparents ^3^	4.969 (2.816–8.767) ***	5.883 (3.404–10.166) ***	7.666 (4.569–12.861) ***	8.279 (4.768–14.377) ***

Odds ratios and 95% confidence intervals of daily informal caregiving at each survey wave were estimated using multivariate binomial logistic regression analyses. A full information maximum likelihood method was used to handle missing data. * Significant at *p* < 0.05; ** *p* < 0.01; *** *p* < 0.001.^1^ Number of siblings when the child was 10 years old was used for all analyses. ^2^ Less than 400 million yen. ^3^ Information on cohabiting with grandparents was not collected at the fourth–wave survey when the child was 16 years old. Thus, cohabiting with grandparents at 14 years of age was used in the analysis of daily informal caregiving at 16 years of age.

**Table 5 ijerph-20-06482-t005:** Longitudinal association between duration of daily informal caregiving from 10 to 16 years of age and household characteristics at 10 years of age.

	Occasional Caregiving ^1^	Persistent Caregiving ^2^
Sex, girl	1.312 (0.928–1.854)	1.729 (0.973–3.071)
Number of siblings	1.101 (0.879–1.381)	1.206 (0.874–1.663)
Single parent	0.802 (0.341–1.890)	1.740 (0.706–4.288)
Low annual household income	1.299 (0.721–2.341)	1.405 (0.648–3.046)
Cohabiting with grandparents	4.754 (3.103–7.284) ***	8.549 (4.684–15.605) ***

Odds ratios and 95% confidence intervals were estimated using multivariate binomial logistic regression analyses with never-daily caregiving as a reference group. A full information maximum likelihood method was used to handle missing data. ^1^ “Occasional” group had daily caregiving at one wave from 10 to 16 years of age. ^2^ “Persistent” group had two or more waves. *** Significant at *p* < 0.001.

## Data Availability

The data that support the findings of this study are available upon request from the last author. The data are not publicly available because of privacy and ethical restrictions.

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
