# Peer review of "Informal Caregiving in Adolescents from 10 to 16 Years Old: A Longitudinal Study Using Data from the Tokyo Teen Cohort"

_ijerph, 2023, doi:10.3390/ijerph20156482_

Round 1

Reviewer 1 Report

The Authors present a paper: " Informal caregiving in adolescents from 10 to 16 years old: a longitudinal study using data from the Tokyo teen cohort" orininal in its content and well written and studied.

I don't have major comments but few suggestions:

1. please give information on the type of used questionnaire

2. please precisize the requested time to fill out the questionnaires

3. is it a self compilation or addressed by somebody?

Author Response

General comment

The Authors present a paper: " Informal caregiving in adolescents from 10 to 16 years old: a longitudinal study using data from the Tokyo teen cohort" orininal in its content and well written and studied.

I don't have major comments but few suggestions:

Response to the general comment

Thank you for your comment. We have incorporated the suggested changes in the manuscript and hope that the revisions sufficiently enhance the quality of the paper.

Comment 1-1

  1. please give information on the type of used questionnaire.

Response to comment 1-1

Thank you for your suggestion. We have used paper questionnaires for first- to third-wave surveys. In the fourth-wave survey, due to the COVID-19 pandemic and related physical distancing measures, we used online questionnaires for data collection. We have added the information on Lines 124–126.

Comment 1-2

  1. please precisize the requested time to fill out the questionnaires.

Response to comment 1-2

Following your suggestion, we have added the following information to the manuscript on Lines 126–127:

Participating children and household respondents took 60 minutes to complete the questionnaire during each wave.

Comment 1-3

  1. is it a self compilation or addressed by somebody?

Response to comment 1-3

Both children and primary household respondents filled out self-administered questionnaires in TTC surveys. However, in this study, all variables used in the analyses were collected by primary household respondents. We have mentioned this information and shown the whole structure of variables in Table 1 on Lines 154–157.

Reviewer 2 Report

Informal Caregiving in Adolescents from 10 to 16 Years Old: A Longitudinal Study Using Data from the Tokyo Teen Cohort

Abstract

-       Please emphasize the study novelty in the abstract

Introduction

-       There is no adequate urgency for the study.

-       Please raise the study novelty.

-       Why the study must be conducted in Tokyo? What makes Tokyo different? What is the impact in Tokyo with informal caregiving on adolescents?

-       The introduction was not well structured so the reader might get a misleading to conclude conclusion of the introduction section. It is better if the authors make a sub-title of the introduction to provide the issue of adolescents, the children, or the informal caregiver itself, or the impact of this issue in Tokyo, etc. It might help the reader comprehend the study introduction.

-       Please state the clear and detailed variables of the study’s aim

Method

-       Please state the clear and detailed variables of the study measurement

-       The variable definition was also important

Result

-       If the model was adjusted with the covariate, this information must be informed well in the result section.

-       No table informs which covariates are significant.

Discussion & conclusion

-       Please provide the main result in the early paragraph of the discussion.

-       Please make the conclusion more detailed.

-       Please provide suggestions to related stakeholders. It might the suggestions for future studies, government, participants or people who are related to the study result, the health department of Tokyo, the social care department, etc.

NA

Author Response

Comment 2-1

Abstract

- Please emphasize the study novelty in the abstract

Response to comment 2-1

Thank you for your suggestion. We have added the following information to the abstract to highlight the novelty of the study on Lines 24–25:

“However, the quantitative evidence regarding household factors relating to informal caregiving has mostly been based on cross-sectional findings.”

Lines 35–36: “Our longitudinal examination highlighted cohabiting with grandparents as a preceding factor for persistent caregiving.”

Comment 2-2

Introduction

- There is no adequate urgency for the study.

Response to comment 2-2

Thank you for the comment. We have added a statement to call for urgent research on factors for young informal caregiving on Lines 77–81:

“Despite growing evidence showing the experiences and outcomes of young informal caregivers, the lack of identification may lead to delayed or missed support delivery through key stages of childhood/adolescence. Hence, there is an urgent need for early identification of young informal caregiving.”

Comment 2-3

- Please raise the study novelty.

Response to comment 2-3

We have added the following information on Lines 96–100:

“Additionally, our longitudinal examination can also illuminate preceding factors by prioritizing considerations for persistent caregiving. Specifically, we featured cohabit-ing with grandparents in addition to sex, number of siblings, single parent, and low annual household income among factors assumed to be associated with informal care-giving [14].”

Comment 2-4

- Why the study must be conducted in Tokyo? What makes Tokyo different? What is the impact in Tokyo with informal caregiving on adolescents?

Response to comment 2-4

Based on our research purpose, we chose to use data from the Tokyo Teen Cohort Study because it is a representative adolescent cohort study in Japan. In Japan, the first nationwide survey on the prevalence of young informal caregivers was administered in 2020. Yet, there has been no national legislation for identification and support for young informal caregivers. The Tokyo metropolitan region has had the largest number of children across 47 regions. The number of children increased in Tokyo over a five-year period from 2015 to 2020, despite a declining trend in all other 46 regions. Population-based sampling from inhabitants in metropolitan areas enables TTC to include a large number of children in the cohort using a prospective design.

We have inserted a third subsection, titled “Tokyo Teen Cohort (TTC),” in the Introduction section to explain this background information (Lines 82–91).

Comment 2-5

- The introduction was not well structured so the reader might get a misleading to conclude conclusion of the introduction section. It is better if the authors make a sub-title of the introduction to provide the issue of adolescents, the children, or the informal caregiver itself, or the impact of this issue in Tokyo, etc. It might help the reader comprehend the study introduction.

Response to comment 2-5

Thank you for the suggestion. We have inserted subtitles in the Introduction section:

1.1. Informal caregiving among children and adolescents

1.2. Challenges in the identification of young informal caregivers

1.3. Tokyo Teen Cohort (TTC)

1.4. Aim of the study

Comment 2-6

- Please state the clear and detailed variables of the study’s aim

Response to comment 2-6

Following your suggestion, we have added the following information:

Lines 98–100:

“Specifically, we featured cohabiting with grandparents in addition to sex, number of siblings, single parent, and low annual household income among factors assumed to be associated with informal caregiving [14].”

Comment 2-7

Method

- Please state the clear and detailed variables of the study measurement

Response to comment 2-7

Following your suggestion, we have provided a detailed explanation of the measurements on Lines 185–196:

“The number of siblings and those cohabiting with grandparents were identified based on responses regarding household composition. Primary household respondents were asked to describe all household members’ sex, age, and relationship (i.e., brother, sister, mother, father, grandmother, grandfather) with the participating children. The number of siblings was calculated by summing the numbers of brothers and sisters. Cohabiting with grandparents was coded with binary categories: “yes” (one or more grandparents in the household) or “no.” Single parent was identified based on the primary household respondent’s response about having a partner or not. Annual household income was assessed using four categories: “less than four million yen,” “4–6.99 million yen,” “7–9.99 million yen,” and “10 or more million yen.” In a multivariate analysis, we re-classified responses into binary categories: “less than four million yen” and “four or more million yen”. Responses were re-classified into binary categories, “less than four million yen” and “four million yen or more,” to perform multivariate analysis.”

Comment 2-8

- The variable definition was also important

Response to comment 2-8

As mentioned in response to comment 2–7, we have provided information regarding the variable definitions.

Comment 2-9

Results

- If the model was adjusted with the covariate, this information must be informed well in the result section.

Response to comment 2-9

We have mentioned that the model was adjusted for all covariates on:

Lines 210–212:

“Independent variables included all covariates at each wave, including the child’s sex, number of siblings, single parent, low annual household income, and cohabiting with grandparents.”

Lines 216–218:

“Independent variables included all covariates comprising the child’s sex, number of siblings, single parent, low annual household income, and cohabiting with grandparents when the child was 10 years old.”

Comment 2-10

- No table informs which covariates are significant.

Response to comment 2-10

We indicated significant coefficients in bold characters in the first manuscript. However, we acknowledge that they could be overlooked. Therefore, we have added asterisks (* p < .05, ** p < .01, *** p < .001) to significant coefficients in the tables in the revised manuscript.

Comment 2-11

Discussion & conclusion

- Please provide the main result in the early paragraph of the discussion.

Response to comment 2-11

Following your feedback, we have added the following information to the discussion section on Lines 293–294:

“Our longitudinal examination highlights cohabiting with grandparents as a preceding factor for persistent caregiving.”

Comment 2-12

- Please make the conclusion more detailed.

Response to comment 2-12

Thank you for the suggestion. We have divided a sentence of the results’ summary into two parts to improve clarity (Lines 346–349):

“Using a large population-based dataset coupled with quantitative methods, this study demonstrated the consistent associations between cohabiting with grandparents and daily caregiving. Furthermore, the longitudinal association implied that cohabiting with grandparents may be a preceding factor for persistent caregiving.”

Comment 2-13

- Please provide suggestions to related stakeholders. It might the suggestions for future studies, government, participants or people who are related to the study result, the health department of Tokyo, the social care department, etc.

Response to comment 2-13

We have suggested marking cohabiting with grandparents as a potential signal of young informal caregiving for social care sectors on Lines 330–334:

“Implications of this study include marking cohabiting with grandparents as a potential signal of young informal caregiving for social care sectors. Future studies that identify the mechanism of persistent caregiving in cohabiting households would provide in-sights into risk factors and prevention. Further policy, regulation, and educational ef-forts should be explored to increase gatekeeping ability in social care service systems for older adults.”

Reviewer 3 Report

The presented research is very interesting because it is not often undertaken. An additional value is their repeatability through four stages of research.

In the characteristics of the research procedures, I missed a more detailed discussion of the research tool (questionnaire) and information whether this tool was verified in terms of reliability.

In lines 108-109 there is information about obtaining permission to conduct research - maybe it would be worth adding the name of the relevant institutions here?

In the discussion of the results, I would suggest adding an analysis and interpretation of the increasing care time resulting, for example, from Table 1.

Author Response

General comment

The presented research is very interesting because it is not often undertaken. An additional value is their repeatability through four stages of research.

Response to the general comment

Thank you for your comments. We have incorporated the suggested changes in the manuscript and hope that the revisions sufficiently enhance the quality of the paper.

Comment 3-1

In the characteristics of the research procedures, I missed a more detailed discussion of the research tool (questionnaire) and information whether this tool was verified in terms of reliability.

Response to comment 3-1

Thank you for your comment. The instruments used in TTC have been developed with input from an international advisory board. However, in the fourth-wave survey during the COVID-19 pandemic, online questionnaires were adapted instead of paper questionnaires that were administered during the first- to third-wave surveys. Therefore, the consistency between paper and online administration should be considered with caution. We have mentioned paper and online administration on Lines 124–126, and discussed the limitations on Lines 339–341.

Comment 3-2

In lines 108-109 there is information about obtaining permission to conduct research - maybe it would be worth adding the name of the relevant institutions here?

Response to comment 3-2

Thank you for the comment. As instructed by the journal regulations, we have described the names of the relevant institutions in the Institutional Review Board Statement on Lines 369–372.

Comment 3-3

In the discussion of the results, I would suggest adding an analysis and interpretation of the increasing care time resulting, for example, from Table 1.

Response to comment 3-3

Following your suggestion, we performed the Kruskal-Wallis H test between four-wave survey to examine whether the frequency of informal caregiving changed over time (age) (Lines 205–206). We found no significant change by age. We have mentioned the results on Lines 229–231 and Lines 287–289.

Reviewer 4 Report

Dear Authors, This is an important area of research.  The results of the study make a clear case for identifying and supporting young carers to ensure they are not disadvantaged.  

I suggest these very minor changes to enhance the readability of a few sentences.   

Line 33, "The mechanism of persistent caregiving needs requires clarification."  Delete the word needs: "The mechanism of persistent caregiving requires clarification."

Line 254, "...information on the person who was cared by the child."  Insert the word for in front of by: "...information on the person who was cared for by the child."

Line 257, "...adults, subletting care for siblings..". Replace the word subletting with the word delegating: "..adults, delegating care for siblings..."

Line 261, "...occurs and forms into persistent care responsibilities.". Replace the words forms into with the word becomes: "..occurs and becomes persistent care responsibilities..."

Line 288, "...young informal caregivers in stakeholders of social care service systems..." Replace of the word in with the word as: "...young informal caregivers as stakeholders of social care service systems..."

High-quality scientific English. Please see the suggest minor changes listed in the box above.

Author Response

General comment

Dear Authors, This is an important area of research. The results of the study make a clear case for identifying and supporting young carers to ensure they are not disadvantaged.

I suggest these very minor changes to enhance the readability of a few sentences.

Response to the general comment

Thank you for your comments. We have incorporated the suggested changes in the manuscript and hope that the revisions sufficiently enhance the quality of the paper.

Comment 4-1

Line 33, "The mechanism of persistent caregiving needs requires clarification." Delete the word needs: "The mechanism of persistent caregiving requires clarification."

Response to comment 4-1

Thank you for your suggestion. We have amended the sentence according to the suggestion (Lines 37–38).

Comment 4-2

Line 254, "...information on the person who was cared by the child."  Insert the word for in front of by: "...information on the person who was cared for by the child."

Response to comment 4-2

We have inserted the term accordingly (Lines 306–307).

Comment 4-3

Line 257, "...adults, subletting care for siblings..". Replace the word subletting with the word delegating: "..adults, delegating care for siblings..."

Response to comment 4-3

We have revised the sentence following your suggestion (Lines 308–310).

Comment 4-4

Line 261, "...occurs and forms into persistent care responsibilities.". Replace the words forms into with the word becomes: "..occurs and becomes persistent care responsibilities..."

Response to comment 4-4

We have changed the sentence following your suggestion (Lines 313–314).

Comment 4-5

Line 288, "...young informal caregivers in stakeholders of social care service systems..." Replace of the word in with the word as: "...young informal caregivers as stakeholders of social care service systems..."

Response to comment 4-5

We have modified the sentence following your suggestion (Lines 349–351).